# The Bioactivity of a Hydroxytyrosol-Enriched Extract Originated after Direct Hydrolysis of Olive Leaves from Greek Cultivars

**DOI:** 10.3390/molecules29020299

**Published:** 2024-01-06

**Authors:** Maria Kourti, Zoi Skaperda, Fotios Tekos, Panagiotis Stathopoulos, Christina Koutra, Alexios Leandros Skaltsounis, Demetrios Kouretas

**Affiliations:** 1Laboratory of Animal Physiology, Department of Biochemistry-Biotechnology, School of Health Sciences, University of Thessaly, 41500 Larissa, Greece; mariakourti@med.uth.gr (M.K.); zoiskap94@gmail.com (Z.S.); ftekos@uth.gr (F.T.); 2Department of Pharmacognosy and Natural Products Chemistry, Faculty of Pharmacy, University of Athens, 15771 Athens, Greece; stathopan@pharm.uoa.gr (P.S.); ckoutral@pharm.uoa.gr (C.K.); skaltsounis@pharm.uoa.gr (A.L.S.)

**Keywords:** olive leaves, hydroxytyrosol, one-step hydrolysis, antioxidant activity, DNA protection

## Abstract

Nowadays, olive leaf polyphenols have been at the center of scientific interest due to their beneficial effects on human health. The most abundant polyphenol in olive leaves is oleuropein. The biological properties of oleuropein are mainly due to the hydroxytyrosol moiety, a drastic catechol group, whose biological activity has been mentioned many times in the literature. Hence, in recent years, many nutritional supplements, food products, and cosmetics enriched in hydroxytyrosol have been developed and marketed, with unexpectedly positive results. However, the concentration levels of hydroxytyrosol in olive leaves are low, as it depends on several agricultural factors. In this study, a rapid and easy methodology for the production of hydroxytyrosol-enriched extracts from olive leaves was described. The proposed method is based on the direct acidic hydrolysis of olive leaves, where the extraction procedure and the hydrolysis of oleuropein are carried out in one step. Furthermore, we tested the in vitro bioactivity of this extract using cell-free and cell-based methods, evaluating its antioxidant and DNA-protective properties. Our results showed that the hydroxytyrosol-enriched extract produced after direct hydrolysis of olive leaves exerted significant in vitro antioxidant and geno-protective activity, and potentially these extracts could have various applications in the pharmaceutical, food, and cosmetic industries.

## 1. Introduction

The cultivation of *Olea europaea* trees characterizes the agricultural economies of Mediterranean countries, including Greece. Annually, large amounts of “olive by-products” are generated from the cultivation of olive trees and olive industrial processing, with limited practical applications [1,2]. The rational exploitation and utilization of agricultural by-products, as well as the production of innovative and quality products in food or other industries, are fundamental conditions for the development and economic reconstruction of the agrifood sector.

Worldwide, an average of 3 tons of olive pruning debris are produced per ha of olive orchard [3], while in Greece, approximately 1,000,000 tons of biomass are produced from olive tree pruning every year [4]. There are two sources for the origin of OLs as olive by-products: (a) the annual olive tree pruning procedure and (b) the olive oil extraction industry. However, OLs in general are either ground and plowed into soil or left on the land for incineration, contributing to serious air pollution [5]. Only a small portion of OLs, compared to the huge amount of disposed materials, are used in the food and pharmaceutical industries or in cosmetics [5]. In this direction, the utilization of olive by-products is at the center of interest to avoid their adverse environmental consequences.

OLs are a natural source of bioactive compounds, including polyphenols, which exert strong antioxidant, antimicrobial, anti-inflammatory, and anti-cancer properties [6]. The main categories of polyphenols in OLs are simple phenols [mainly hydroxytyrosol, (HT)], flavonoids, and secoiridoids (mainly oleuropein) [7]. These secondary metabolites have an ideal chemical composition for acting as free radical scavengers, exerting their antioxidant activity [8,9,10]. However, there are numerous genetic and environmental factors that determine the quantity, composition, and bioavailability of polyphenols in OLs [11].

The most abundant polyphenol in OLs is oleuropein (OLE), which is responsible for the characteristic bitter taste of olives [11]. OLE is a derivative of elenolic acid that is connected to the orthodiphenol HT by an ester bond and to a glucose molecule by a glycosidic bond [12,13]. The hydrolysis of these bonds leads to the separate release of the three molecules [14]. Hydrolysis of OLE can be carried out by enzymatic, alkaline, or acidic methods [14]. Acidic hydrolysis is the most common method used in laboratories and industrial processes [15,16]. The plethora of biological properties of OLE are mainly due to the HT moiety, a drastic catechol group.

HT is a small molecule, so the ratio of its bioactivity to weight is greater than OLE’s. HT exerts strong antioxidant activity and is strongly associated with beneficial effects in various human diseases [17], manifesting anti-tumor [18,19], cardioprotective [20,21], and neuroprotective effects [22,23]. The strong protective activity of HT is also displayed against heavy metal-caused damage. More specifically, HT prevented mercury-induced oxidative stress at red blood cells (RBCs) [24], as well as protecting the rat brain from arsenic (As)-induced oxidative stress [25]. HΤ is generally recognized as safe (GRAS) by the European Food Safety Authority (EFSA) and the Food and Drug Administration (FDA) and is asserted to improve human health when consumed at a minimum dose of 5 mg per day, as described in EU Regulation 432/2012 [26]. Moreover, recently, the European Union, by decision (Regulation 2017/2373 of 14 December 2017), approved the use of HT on the market as a novel food ingredient in accordance with Regulation of the European Parliament (EC) No. 258/97 [27]. Therefore, the use of HT as an additive in food products creates new opportunities for revalorizing olive by-products, rich in this compound [28].

As previously mentioned, the levels of biophenols in OLs depended on several factors, including the olive tree variety, the type of cultivation (organic and conventional), the agronomic practices, and the storage conditions [11,29,30]. The activity of endogenous enzymes (polyphenol oxidase, peroxidase, β-glucosidase, and esterase), which convert OLE to HT, can also determine the levels of these phenols in OLs [31,32]. On the other hand, HT may also be synthesized chemically [33,34,35,36]. However, the cost of the reaction steps and the precursors significantly increases the product price. Since HT is derived from OLE, the most abundant polyphenol in OLs, it is obvious that this by-product could be an ideal raw material for the production of HT-enriched extracts. To our knowledge, most of the protocols that have been proposed in the literature regarding the hydrolysis of OLE include two main steps. Firstly, OLE is extracted from dry OLs, and secondly, it is hydrolyzed to obtain HT under acidic conditions [37,38,39,40]. However, there is not available data for a HT-enriched extract produced by direct hydrolysis of dry OLs and its bioactivity.

The main goal of the present study was to investigate, holistically, the antioxidant and geno-protective effects of an HT-enriched extract originated from the direct hydrolysis of dried OLs (DHOLE) of Greek *Olea europaea* cultivars. Even though there are studies that have mentioned the in vitro antioxidant activity of OL extracts, less is known about hydrolyzed extracts as well as their bioactivity. In this study, we performed a rapid methodology for the DHOLE production from direct acidic hydrolysis of dried OLs, where the extraction procedure and the hydrolysis of OLE are carried out in one step. Subsequently, we explored the possible antioxidant and geno-protective activity of the DHOLE using cell-free and cell-based methods. Our findings could be exploited in order to use the HT-enriched extracts originated from the direct hydrolysis of OLs in many nutritional supplements, food products, and cosmetics due to their positive biological impact.

## 2. Results

### 2.1. Characterization of OLs Extracts

Figure 1 shows the HPLC-DAD chromatograms obtained from the analysis of the methanolic extract of OLs, the extract originated from the hydrolysis of methanolic extract (HOLE), and the extract produced by the direct hydrolysis of dried OLs (DHOLE). The main metabolites in the methanolic and hydrolyzed extracts were OLE and HT, respectively. The content of OLE in the methanolic extract was 9.36%, while a similar content of HT was detected in the HOLE (11.91%) and DHOLE (11.27%) (Table 1).

### 2.2. Assessment of the In Vitro Bioactivity of DHOLE

According to the aforementioned results, the two hydrolyzed extracts, HOLE and DHOLE, had similar HT content. For this purpose, we investigated the in vitro bioactivity of DHOLE using cell-free and cell-based methods, since there is not enough available data about the bioactivity of an extract that originated after direct hydrolysis of OLs.

#### 2.2.1. In Vitro Antioxidant Activity Using Cell-Free Assays

Table 2 summarizes the IC_50_ values of the DHOLE, which exerts strong anti-radical activity in all the tested assays, as the IC_50_ values were lower than (O_2_^•−^, ROO^•^) or close to (DPPH^•^, ABTS^•+^, RP) IC_50_ values of positive controls. It is known that the lower the IC_50_ value, the higher the antioxidant activity. Remarkable is the fact that the extract not only neutralizes the artificial radicals DPPH^•^ [IC_50_: 21.3 μg/mL vs. Ascorbic Acid (AA) IC_50_: 3.8 μg/mL] and ABTS^•+^ (IC_50_: 6.54 μg/mL vs. AA IC_50_: 2.8 μg/mL), but also has a very high ability to neutralize superoxide radicals (O_2_^•−^) (IC_50_: 161.0 μg/mL vs. Ellagic Acid IC_50_: 260.0 μg/mL), which play a crucial role for cell signaling and survival, but in high levels can promote DNA damage, leading to various diseases [41]. Furthermore, the DHOLE exerted a strong ability to reduce iron Fe (III) into Fe (II) (AU_0.5_: 7.9 μg/mL vs. AA AU_0.5_: 5.0 μg/mL), which further reinforced the antioxidant ability of the extract. Finally, the DHOLE exhibited protective activity against ROO^•^-induced DNA plasmid breakage, with a much lower IC_50_ value (IC_50_: 36.4 μg/mL) than the positive control (AA IC_50_: 290.0 μg/mL).

#### 2.2.2. Antioxidant Activity Using Cell-Based Methods

The antioxidant activity of DHOLE at the cellular level was estimated in EA.hy926 and C2C12 cells. These cell lines were used as typical cell models for evaluation of the impact of oxidative stress on vascular endothelium and muscle progenitors’ development and differentiation, respectively.

##### Estimation of DHOLE Cytotoxicity

In order to examine the antioxidant activity of DHOLE in cells, its non-cytotoxic concentrations were defined [42] using the XTT assay after 24 h of treatment at increasing concentrations of DHOLE in serum-free medium (Figure 2). The concentration of DHOLE, where a statistically significant decrease was observed, was defined as cytotoxic (Figure 2). According to the results, the non-cytotoxic concentrations of the extract for EA.hy926 cells (Figure 2a) were 10–80 μg/mL and 5–40 μg/mL for C2C12 cells (Figure 2b), which were used at the following experiments.

##### Assessment of Antioxidant Biomarkers after Treatment with DHOLE

Next, ROS, GSH, and TBARS levels were measured as well-established biomarkers for cellular redox status [42]. ROS and GSH levels were measured using flow cytometry after 24 h of cell treatment without (control) or with 10–80 μg/mL and 5–40 μg/mL of DHOLE, respectively, in serum-free medium. Data showed that ROS levels were statistically significantly increased only at endothelial cells at 20–80 μg/mL DHOLE, while at C2C12 cells there was no alteration. However, GSH levels were not changed in both cell lines (Figure 3). Concerning lipid peroxidation levels, interestingly, a different response was observed between the two cell types. At C2C12 cells, TBARS levels were statistically significantly decreased at lower concentrations of DHOLE (5–20 μg/mL), while a 2-fold increase was mentioned at the higher concentration (40 μg/mL), a finding that suggests pro-oxidant activity. However, at EA.hy926 cells, no statistically significant alteration was noticed (Figure 3).

#### 2.2.3. Geno-Protective Activity of DHOLE

We also investigated the ability of DHOLE to protect DNA from DNA damage under oxidative conditions using two different in vitro approaches, using cell-free and cell-based methods.

##### Protective Activity against ROO^•^-Induced DNA Plasmid Breakage

In the first case, we examined the influence of the presence of DHOLE increasing concentrations (6.25–200 μg/mL) on ROO^•^-induced DNA plasmid breakage (Figure 4). The DNA plasmid protective ability of DHOLE was expressed as an IC_50_ value (IC_50_: 36.4 μg/mL, Table 2), which was much lower than the positive control (AA IC_50_: 290 μg/mL), indicating strong in vitro geno-protective activity (Table 2). This result came from the statistical analysis of the densitometric analysis of the agarose gel data from three independent experiments. The measurements used in order to calculate the inhibition of the conversion of supercoiled conformation (SC) to open circular forms (OC) (Figure 4).

##### Cellular DNA-Protective Effect of DHOLE under Oxidative Conditions

Next, we investigated if the aforementioned in vitro DNA-protective ability of the extract manifested at the cellular level using a Comet assay. For this purpose, EA.hy926 and C2C12 cells were pre-treated with certain DHOLE concentrations for 23 h, followed by the addition or not of 250 μΜ H_2_O_2_ for 1 h as an oxidative agent. The concentrations of the extract for each cell type were chosen based on the results of ROS, GSH, and TBARS measurements, where a positive or no effect was observed at a minimum concentration of the extract. Based on this, EA.hy926 cells were pre-treated with 10 μg/mL of DHOLE, but no effect at the ROS, GSH, and TBARS levers was detected. Similarly, C2C12 cells pre-treated with 5 μg/mL and 10 μg/mL of DHOLE showed decreased TBARS levels, indicating reduced levels of lipid peroxidation in the presence of DHOLE. However, at EA.hy926 cells, we also tested a concentration of DHOLE (40 μg/mL), where increased ROS levels were observed, in order to examine its influence on nuclear DNA.

Figure 5 shows representative images of the Comet assay, where comet tails indicate DNA damage. Untreated cells (control) showed no tails in both cell lines (Figure 5A,G). In contrast, treatment with 250 μM H_2_O_2_ for 1 h caused significant DNA damage, as evidenced by the comet tails in both cell lines (Figure 5B,H). Interestingly, this damage was prevented when cells were pre-treated with DHOLE at lower concentrations (Figure 5D,J) and at higher concentrations (Figure 5F,L). Furthermore, treatment with the extract alone did not affect both cell lines at any concentration (EA.hy926: Figure 5C,E, and C2C12 cells: Figure 5I,K).

The automatic determination of the three parameters TM, TL, and TD of 100 randomly selected cells/sample from Comet images verified the aforementioned observations (Figure 6). In particular, according to the three parameters, the H_2_O_2_ treatment induced high DNA damage in both cell lines (Figure 6), with slightly higher DNA damage in myoblasts (Figure 6B and Appendix A) than in endothelial cells (Figure 6A and Appendix A). However, this damage was significantly reduced in the presence of the extract in both cell lines (Figure 6A,B, and Appendix A). Surprisingly, the extract exerted a stronger DNA-protective effect at lower concentrations in both cell lines (EA.hy926: 10 μg/mL; C2C12: 5 μg/mL), while this protective effect was fading at higher concentrations (EA.hy926: 40 μg/mL; C2C12: 10 μg/mL). In conclusion, pre-treatment with DHOLE at low concentrations, 5 μg/mL for myoblasts and 10 μg/mL for endothelial cells, protected better from DNA damage under oxidative conditions.

## 3. Discussion

The present study demonstrates a holistic approach to the in vitro antioxidant capacity of an extract that originated from the direct hydrolysis of OLs from Greek cultivars, as well as its ability to protect against oxidative DNA damage.

Since HT is a very well-studied component of olive trees with multifaceted biological activity, the production of OL extracts rich in HT has been at the center of interest in the scientific community as well as in food, pharmaceutical, and cosmetic companies because of the high demand in the market for products enriched with natural antioxidants. A frequently cited methodology that has been proposed in the literature to produce HT-rich extracts is acidic hydrolysis. According to this methodology, the hydrolysis of Oleuropein is carried out in a two-step procedure. In the first step, OLE is extracted from dried OLs, and in the second step, it is hydrolyzed to obtain HT under acidic conditions. Recently, Papageorgiou et al. [16] explored the optimal conditions for producing an extract rich in HT from OLs. Specifically, various parameters were studied, including the particle size of plant material, liquid-to-material ratio (g/L), extraction solvent, the impact of temperature during solid–liquid extraction, temperature and duration of hydrolysis, hydrolysis medium (acidic or alkaline hydrolysis), and the type of catalyst (H_2_SO_4_ or HCl), as well as the pH of the aqueous phase before liquid–liquid extraction with ethyl acetate. The findings of this study led to the development of the reference protocol employed in this work for producing the hydrolyzed OLs extract. The only exception was the extraction solvent, where instead of the mixture of solvents ethanol/water, methanol was used for the extraction of OLs, according to the European Pharmacopeia protocol [43]. Specifically, in the present study, two different OLE hydrolysis protocols were applied to OLs, aiming to produce an extract rich in HT. Protocol A refers to the hydrolysis of OLE in the extract and was used as a reference protocol, while protocol B describes the direct hydrolysis of OLE (DHOLE) in OLs. Evaluating the results of the chemical analyses performed on the hydrolyzed extracts (protocols A and B), we found that the yields of the extracts and their HT content were at similar levels.

Applying protocol A, an extract with a yield of 5.30% containing 11.91% HT was produced, while the treatment of olive leaves with protocol B led to the production of an extract with a yield of 5.17% and a HT content of 11.27%. Similar results were obtained by Wang et al. [44], where using the technique of ultrasound, hydrochloric acid as hydrolysis medium, and macroporous resin for the enrichment of HT, an OLs extract containing 9.25% HΤ was produced. At this point, it is worth noting that, according to the results of the HPLC-DAD analysis, the OLE compound was not detected in the hydrolyzed extracts, probably due to its quantitative hydrolysis to HT occurring under acidic conditions. Considering, the content of OLE in dried OLs (2.44 g/100 g) as well as the ratio of the molecular weight of OLE to the molecular weight of HT: 3.52/1, the theoretical amount of HT corresponding to the OL sample is 695.0 mg/100 g. Moreover, based on the results obtained from the analysis of the hydrolyzed extracts, the amount of HT in dried OLs is 631.2 mg/100 g applying protocol A and 582.6 mg/100 g applying protocol B. These data demonstrate that the hydrolysis of OLE, applying protocols A and B, was achieved at a rate of 90% and 85%, respectively.

Afterwards, the extract obtained from the direct acidic hydrolysis of dried olive leaves (DHOLE) was studied further by in vitro biological assays. The in vitro antioxidant activity of DHOLE is estimated using free radical scavenging assays. Comparing DPPH^•^ and ABTS^•+^ assays, DHOLE exhibited a lower IC_50_ value in ABTS^•+^ (IC_50_: 6.54 μg/mL) than in the DPPH^•^ assay (IC_50_: 21.3 μg/mL), which means stronger ability to scavenge ABTS^•+^ radicals. This result can be explained by the different solvents used in the two assays [45], as in the ABTS^•+^ assay the solvent is water, while in the DPPH assay the solvent is MeOH [46]. As a result, the DHOLE, which contains HT, a highly hydrophilic component [47], exerted higher scavenging activity in the ABTS^•+^ assay. This change has also been observed in other previous studies that investigated olive extracts, which are rich in polar molecules [48,49]. With these data, we highlighted the antioxidant capacity of the extract using another two in vitro assays, O_2_^●−^ and RP. DHOLE also had the ability to scavenge superoxide radicals (O_2_^•−^) (IC_50_: 161.0 μg/mL vs. Ellagic Acid IC_50_: 260.0 μg/mL), an essential signaling molecule for the life of aerobic organisms, but at toxic levels could cause DNA damage [50,51].

Despite the fact that there are many studies concerning the in vitro antioxidant activity of OLs extracts [52,53,54,55,56], less is known about hydrolyzed extracts. In the first case, the bioactivity of the extracts is mainly exerted due to OLE, as the main phenol in OLs, while after hydrolysis, the main drastic component is HT. According to a study on extracts after enzymatic hydrolysis of olive leaf extracts, the antioxidant activity using the DPPH^•^ assay was worse than the activity of the extracts before the process, but after ethyl acetate extraction, it was better [57]. The authors proposed that this difference can be explained because the OLE was not completely transformed into HT. Comparing our data using the same method, DHOLE had stronger antioxidant activity (DPPH^•^IC_50_: 21.3 μg/mL), probably due to the better effectiveness of OLE hydrolysis into HT. However, there are various reasons for this difference, such as the quality of olive leaves, storage conditions, or cultivar.

As far as we know, the available studies mainly investigate the bioactivity of hydrolyzed OL extracts in cells using cytotoxicity assays, while there is no data to support the geno-protective effects of a HOLE or DHOLE. However, the antioxidant and DNA-protective activities have been assessed for the isolated bioactive phenols, HT and OLE, separately in different cell types [58].

In the present study, the results for ROS and lipid peroxidation levels showed a cell-type-specific response. In particular, ROS increased levels observed at higher concentrations of DHOLE in endothelial cells, while myoblasts were not affected. On the other hand, decreased TBARS levels were noticed at lower DHOLE concentrations at myoblasts, while no alteration was observed at endothelial cells. The increased ROS levels could be justified by the fact that polyphenols can exert pro-oxidant activity after reaching a crucial concentration, at which point they are converted into radicals [59,60,61]. However, this alteration did not affect TBARS levels (Figure 2) or cellular DNA, as shown by the Comet assay experiments, at endothelial cells (Figure 5 and Figure 6). However, there are other biomolecules and molecular pathways regulated by HT in endothelial functioning. Data have shown that HT increases the levels of nuclear factor-E2-related factor 2 (Nrf2) in endothelial cells [62], which is the key transcription factor for antioxidant defense [63]. HT also protected endothelial cells from H_2_O_2_ by increasing Nrf2 expression and activating the Akt and ERK1/2 pathways [64]. However, it is not clear the exact molecular mechanism by which HT can lead to geno-protection.

An interesting finding was the sharp rise of TBARS levels at the highest DHOLE concentration at myoblasts (Figure 2). This result may be explained by the potential pro-oxidant activity of polyphenols after reaching a ‘crucial’ concentration and by the way that they are metabolized [65]. The pro-oxidant effect of high concentrations of HT has also been confirmed by in vivo experiments in exercised rats [66]. These results were accompanied by no change in GSH levels in both cell lines. In summary, DHOLE antioxidant activity at the cellular level was exerted by reducing TBARS levels at myoblasts, but it did not induce any antioxidant markers at endothelial cells.

On the other hand, DHOLE exerted strong DNA-protective activity under oxidative conditions using cell-free and cell-based methods. In the first case, DHOLE protected from plasmid breakage induced by ROO^•^, with an IC_50_ value much lower (IC_50_: 36.4 μg/mL) than ascorbic acid (positive control) (IC_50_ AA: 290 μg/mL). The geno-protective ability of DHOLE was also verified at the cellular level using an alkaline Comet assay. This method is a useful tool for estimating DNA damage and repair, detecting single-stranded, double-stranded DNA breaks, and alkali-labile sites [67,68]. We found that DHOLE pre-treatment protected from H_2_O_2_-induced DNA damage in both cell lines, as comet tails were significantly reduced or absent in the presence of extract in comparison to H_2_O_2_-treated cells (Figure 5 and Figure 6). Interesting is the fact that this geno-protective effect was stronger at lower concentrations of DHOLE than at higher concentrations in both cell lines (Figure 5 and Figure 6). This geno-protective ability of DHOLE could be attributed to HT activity, the main phenolic compound of DHOLE, which can act as a free radical scavenger, activator of DNA repair, and antioxidant defense mechanism [58]. Our results are in agreement with another study in which HT exerted an anti-apoptotic effect on C2C12 cells treated with H_2_O_2_ as the apoptotic agent [69].

Considering the aforementioned data, we conclude that DHOLE is bioactive and exerts strong in vitro antioxidant and geno-protective effects using cell-free and cell-based methods. The novel one-step methodology of direct hydrolysis of dried OLs that was used is a rapid and effective way for the production of HT-enriched extracts from OLs, which could have various applications in the food industry or in the pharmaceutical industry.

## 4. Materials and Methods

### 4.1. Production of Enriched-in-Hydroxytyrosol Extract from Olive (Olea europaea) Leaves

All chemicals for the following assays were purchased from Sigma-Aldrich, Munich, Germany, except for the HT and OLE reference standards, which were purchased from ExtraSynthase (Lyon Nord, France).

#### 4.1.1. Preparation of the Raw Material

The OLs used in this work originated from the University Campus of NKUA and were collected in February 2022. Specifically, small branches were gathered during the period of tree pruning and placed in a dry, protected from light place for 4 weeks, until most of their moisture was removed. After this period, the residual humidity of the OLs was less than 5%, which was sufficient for their pulverization with a standard laboratory blender. Olive powder was then sieved into particles smaller than 0.71 mm and stored at 8–10 °C prior to extraction.

#### 4.1.2. Protocol (A): A Two-Step Procedure for the Production of a HT-Enriched Extract (HOLE)

The recovery of OLE from OLs was carried out using methanol as an extraction solvent. Specifically, 10 g of the dried powdered leaves were mixed with 100 mL of methanol, and then the mixture was left in a silica oil bath at 60 °C for 30 min, under stirring. Afterwards, the solvent was evaporated to dryness under a water pump vacuum. The % *w*/*w* yield of the extract in dried OLs was 26.14%.

The hydrolysis of OLE to HT was performed in the extract of dried OLs using a 2 M H_2_SO_4_ aqueous solution, taking into consideration the optimal conditions proposed by Papageorgiou et al. [16]. Specifically, the obtained methanolic extract was mixed with 100 mL of 2 M H_2_SO_4_, and then the mixture was left in a silica oil bath at 50 °C for 30 min, under stirring. The hydrolysis reaction was stopped by immersing the reaction vessel in an ice bath, and then an aqueous of 3 M sodium hydroxide (NaOH) solution was used to neutralize the hydrolysate and adjust the pH to a range of 4–5. After that, the neutralized solution was filtered, and HT was recovered from the hydrolysate by liquid–liquid extraction with ethyl acetate. For this purpose, three equal portions of ethyl acetate were introduced in the neutralized solution, with each portion being one third in volume compared to the aqueous phase. The % *w*/*w* yield of HOLE in dried OLs was 5.30%. After that, an aliquot of the dried extract was diluted in a mixture solvent of methanol/waterː 1/1 (*v*/*v*) and forwarded for HPLC-DAD analysis.

#### 4.1.3. Protocol (B): One-Step Procedure for the Production of a HT-Enriched Extract (DHOLE)

The recovery of OLE and its hydrolysis to HΤ were carried out on dried OLs in a one-step process using a 2 M H_2_SO_4_ aqueous solution. Specifically, 10 g of the dried powdered leaves were mixed with 100 mL of 2 M H_2_SO_4_, and then the mixture was left in a silica oil bath at 85 °C for 60 min, under stirring. The reaction was stopped by immersing the reaction vessel in a cool bath, and an aqueous solution of 3 M NaOH was used to neutralize the hydrolysate and adjust the pH to a value range of 4–5. After that, the neutralized solution was filtered, and HT was recovered from the hydrolysate by liquid–liquid extraction with ethyl acetate, following the procedure described previously. The % *w*/*w* yield of DHOLE in dried OLs was 5.17%. Afterwards, an aliquot of the dried extract was diluted in a mixture solvent of methanol/waterː 1/1 (*v*/*v*) and forwarded for HPLC-DAD analysis.

#### 4.1.4. HPLC-DAD Analysis and Quantification

Olive biophenols were determined in both hydrolyzed and non-hydrolyzed OL extracts by applying the IOC-proposed analytical method with some modifications. The IOC method was performed according to analytical conditions referred to in the IOC/T.20/Doc No. 29 method (International Olive Council, 2009) [70], and they have been described in our previous work [48]. Specifically, the separation of the components of the extracts was achieved on a reversed-phase Spherisorb Discovery HS C18 column (250 × 4.6 mm, 5 μm; Supelco, Bellefonte, PA, USA) using a mobile phase consisting of 0.2% aqueous orthophosphoric acid (A) and MeOH/ACN (acetonitrile) (50:50 *v*/*v*) (B), at a flow rate of 1.0 mL/min and ambient temperature. The applied gradient elution was as follows: 0 min, 96% A and 4% B; 40 min, 50% A and 50% B; 45 min, 40% A and 60% B; 60 min, 0% A and 100% B; 70 min, 0% A and 100% B; 72 min, 96% A and 4% B; 82 min, 96% A and 4% B. The injection volume was held constant at 20 μL, and chromatograms were monitored at 280 nm. All analyses were made in triplicate. The determination of the main phenolic compounds in OLs extracts was achieved using the regression analysis method. Specifically, standard calibration curves for HT and OLE were prepared. For the HT quantification, 10-point calibration curves were constructed (HT: y = 87594x + 31144, r^2^ = 0.999), while OLE quantified according to a 6-point calibration curve, respectively (OLE: y = 22405x + 313038, r^2^ = 0.998).

### 4.2. In Vitro Cell-Free Methods

All chemicals for the following assays were purchased from Sigma-Aldrich, Munich, Germany.

#### 4.2.1. DPPH^•^ Radical Scavenging Assay

The DPPH^•^ assay was evaluated as described previously [49,71]. Briefly, a methanolic dilution of 100 μM DPPH^•^ radical was mixed with increasing concentrations of the extract in a total volume of 1.0 mL in triplicate. After a 20 min incubation of the samples in the dark at room temperature (RT), the absorbance was measured at 517 nm on a Hitachi U-1900 radio beam spectrophotometer (serial No. 2023-029; Hitachi, Tokyo, Japan). In each experiment, methanol by itself served as the blank, and the DPPH radical by itself in methanol served as the control. The % radical scavenging capacity (RSC) was then determined using the equation below:RSC (%) = (Acontrol − Asample)/Acontrol × 100
where Acontrol and Asample are the absorbance values of the control and the test sample, respectively. Next, from the graph-plotted %RSC versus the sample concentration, the IC_50_ value was calculated, which is the concentration of the extract that causes 50% scavenging of DPPH^•^ radical. As a positive control, we used ascorbic acid.

#### 4.2.2. ABTS^•+^ Radical Scavenging Assay

The 2,2′-Azino-bis-(3-ethyl-benzthiazoline-sulphonic acid) (ABTS) radical cation (ABTS^•+^) decolorization assay was determined according to the method described by Cano et al. (1998) [72], with some modifications [49,71]. In brief, in each reaction tube, in triplicate, were added the following: 500 μL of ABTS (1 mM), 50 μL H_2_O_2_ (30 μM), 50 μL horseradish peroxidase (6 μM), and 400 μL dionized water (dH_2_O). After a 45 min incubation of the reaction tubes in the dark at RT, 50 μL of increasing concentrations of DHOLE were added to the reaction mixture. Finally, the absorbance was measured at 730 nm (serial No. 2023-029; Hitachi). The blank sample did not contain the peroxidase, while the control samples did not contain the extract. The RSC (%) and the IC_50_ values were estimated in the same manner as described in the DPPH^•^ assay. Ascorbic acid was used as a positive control.

#### 4.2.3. Superoxide (O_2_^•−^) Radical Scavenging Assay

Superoxide (O_2_^•−^) radical scavenging activity was determined according to the method of Gülçin et al. (2004) [73]. Briefly, 625 μL of Tris–HCl (16 mM, pH 8.0), 125 μL of NBT (100 μM), 125 μL of NADH (468 μM), and 50 μL of extract at increasing concentrations were mixed in triplicate. Moreover, 125 μL of phenazine methosulfate (PMS) (60 μM) was added, and the tested tubes were incubated for 5 min and centrifuged at 3000 rpm for 10 min at 25 °C. Next, the absorbance was measured at 560 nm (serial No. 2023-029; Hitachi). In each experiment, the samples without PMS were used as blanks, and the samples without extract were used as controls. The RSC (%) and the IC_50_ values were determined as described above for the DPPH^•^ assay. Instead of ascorbic acid, ellagic acid was used as a positive control.

#### 4.2.4. Reducing Power Capacity

The assay was performed according to the protocol of Yen and Duh (1993) [74], with some modifications [71]. In brief, 200 μL of phosphate buffer (0.2 M, pH 6.6) and 250 μL of potassium ferricyanide (1% *w*/*v*) were mixed with 50 μL of increasing concentrations of brine extract in triplicate. Then, the reaction tubes were incubated at 50 °C for 20 min and cooled on ice for 5 min. Next, 250 μL of TCA (10% *w*/*v*) was added, and the samples were centrifuged at 3000 rpm for 10 min at 25 °C. Following, reaction tubes containing 700 μL of the supernatant, 250 μL of deionized water, and 50 μL ferric chloride (0.1% *w*/*v*) were incubated for 10 min at RT. The results are expressed as the AU_0.5_ value, which was calculated from the graph-plotted absorbance against the extract concentration, indicating the extract concentration that causes an absorbance of 0.5 at 700 nm. Ascorbic acid was used as a positive control.

#### 4.2.5. Peroxyl Radical-Induced DNA Strand Cleavage Assay

The assay was performed according to a previously described protocol [75] with some modifications [76]. Briefly, in order to cause DNA plasmid breakage, 3.2 μg pBluescript-SK+ plasmid DNA was treated with 95 mM 2,2′-azobis (2-amidinopropane hydrochloride) (AAPH) in PBS in the dark for 45 min at 37 °C. Next, 3 μL of loading buffer was added to each sample, and electrophoresis on a 0.8% *w*/*v* agarose gel was performed at 80 V for 55 min. After ethidium bromide staining, gels were exposed to UV, and images were taken using a MultiImage Light Cabinet and analyzed by Alpha view suite (version: 2.0.0.9) image analysis quantification software.

The ability of DHOLE to protect from ROO^•^-induced plasmid strand breakage was estimated by measuring the inhibition of the conversion of supercoiled conformation to open circular and linear forms according to the following equation:% inhibition = (S − So)/(Scontrol − So) × 100
where Scontrol: % supercoiled DNA in plasmid DNA alone (negative control), So: % supercoiled DNA in the plasmid DNA with AAPH alone (positive control), and S: % supercoiled DNA in the presence of the tested extract and AAPH. The results are presented in a graph plot of the %inhibition versus the extract concentration, from which IC_50_ values were estimated, indicating the concentration needed to inhibit relaxation of the supercoiled conformation induced by peroxyl radicals by 50%. The experiment was carried out three independent times. Ascorbic acid was used as a positive control.

### 4.3. Cell-Based Assays

#### 4.3.1. Cell Culture

EA.hy926 endothelial cells were cultured in 1 g/L glucose, while C2C12 murine myoblasts were cultured in 4.5 g/L glucose Dulbecco’s modified Eagle’s medium (DMEM), in the addition of 10% (*v*/*v*) fetal bovine serum (FBS) and 100 U/mL of penicillin/streptomycin, at 37 °C in 5% CO_2_ [49]. All cell culture materials were purchased from Gibco, Thermo Fisher Scientific, Waltham, MA, USA.

#### 4.3.2. XTT Assay

The cytotoxicity of DHOLE was assessed using the XTT assay kit (Roche, Mannheim, Germany), according to the manufacturer’s instructions. Briefly, 10 × 10^3^ EA.hy926 cells and 5 × 10^3^ C2C12 cells per well were seeded into a 96-well plate in complete medium. The next day, the complete medium was removed, and 100 μL of increasing concentrations of DHOLE in serum-free DMEM were applied. After a 24 h incubation period, 50 μL of XTT test solution were added to each well and incubated for 4 h. Next, the absorbance was measured at 450 nm and at 630 nm as a reference wavelength in a microplate reader (BioTek Instruments, Inc., Winooski, VT, USA). Serum-free DMEM alone and in the presence of increasing concentrations of DHOLE were used as a negative control, which were subtracted from the corresponding incubations in the cells. The % of cell proliferation was calculated according to the following equation:Cell proliferation (%) = [(ODcontrol − ODsample)/ODcontrol] × 100,
where ODcontrol and ODsample indicate the OD of untreated and treated cells, respectively. For each cell line, at least three independent experiments were carried out in triplicate.

#### 4.3.3. Flow Cytometry for ROS and GSH Detection

Flow cytometry was used for the detection of the intracellular ROS and GSH levels using the DCF-DA and intracellular glutathione (GSH) Detection Assay Kit (ab112132; Abcam, Cambridge, UK), respectively. Furthermore, 200,000 cells/well were seeded in two separate 6-well plates, one for ROS and one for GSH detection. Cells were treated in the presence of increasing concentrations of the extract for 24 h in serum-free medium. After the treatment, ROS detection was performed by placing 1 mL of 10 μΜ DCF-DA (20 mM stock in DMSO) in PBS in each well after incubation for 45 min at 37 °C. After this step, the cells were trypsinized, centrifuged (1200 rpm, 5 min, 4 °C), and re-suspended in 250 μL of PBS. Simultaneously, GSH detection was performed according to the manufacturer’s instructions. Briefly, cells were trypsinized and re-suspended in 1 mL of PBS with 5 μL of green dye and incubated for 30 min at 37 °C in the dark. Next, cells were centrifuged (1200 rpm, 5 min, 4 °C) and re-suspended in 250 μL of PBS. Then, all samples were submitted to flow cytometric analysis using a FACScan flow cytometer (Becton Dickinson, NJ, USA) with excitation and emission length at 490/520 nm for ROS and for GSH detection.

#### 4.3.4. Lipid Peroxidation Levels

After 24 h of treatment with or without increasing concentrations of DHOLE in serum-free medium, cells were washed with PBS and scraped from 25-cm^3^ flasks in 250 μL of PBS, containing a Roche cOmpleteTM protease inhibitor cocktail tablet (Roche Diagnostics, Mannheim, Germany). The cells were then periodically ultrasonically stimulated (70% amplitude, 0.5 s pulse cycle) on ice for 10 s, followed by a 10 s break (UP400S, Hielscher, Teltow, Germany). This process was performed a total of five times. Following a centrifugation (15,000× *g*, 20 min, 4 °C), the supernatant was gathered. Here, 70 μg of total protein were utilized for each sample, in duplicate, to determine the protein concentration using the Bradford assay, which used a standard curve of bovine serum albumin as a reference. Tris-HCl (200 mM, pH = 7.4), 500 mL of 35% TCA, and 400 μL of PBS with or without cell lysate (blank) were all added, and the samples were incubated for 10 min at room temperature. Afterward, 1 mL of 2 M Na_2_SO_4_-55 mM TBA buffer was added, and the samples were then heated in a water bath for 45 min at 95 °C. Following incubation, the samples were centrifuged at 11,200× *g* for 3 min at 25 °C, after cooling on ice for 5 min, and 1 mL of 70% TCA was added. The absorbance was subsequently determined at 530 nm. Malondialdehyde (MDA) (156,000 L/mol/cm) was used to compute the TBARS levels, and the findings were displayed as nmol MDA/mg protein.

#### 4.3.5. Alkaline Comet Assay

The alkaline Comet assay was performed as previously described [48]. Briefly, 2 × 10^5^ cells/well were pre-treated at specific concentrations of DHOLE for 24 h. After a 23 h treatment, 250 µM H_2_O_2_ was added for 1 h in the presence or absence of the extract. Then, the cells were trypsinized, and a cell suspension of 1 × 10^6^ cells/mL in PBS was prepared. Then, on pre-coated microscope slides with 1% *w*/*v* normal-melting agarose, a mixture of 20 μL cell suspension and 80 μL of pre-warmed 0.5% *w*/*v* low-melting agarose in PBS were layered. The lysis step was performed in pre-cooled lysis buffer (2.5 Μ NaCl, 100 mM EDTA, 10 mM Tris base, NaOH was added to pH 10.0, and 1% Triton X-100 was freshly added) at 4 °C for 2 h. DNA unwinding was performed in cold-fresh electrophoresis buffer (300 mM ΝaOH, 1 mM EDTA, pH > 13) for 2 × 10 min at 4 °C, and electrophoresis (25 V/300 mA) was performed at 4 °C for 20 min. After that, the neutralization of samples was performed in a cold neutralization buffer (0.4 M Tris–HCl, pH 7.5) for 2 × 15 min each time. Then, the slides were washed once in dH_2_O for 5 min, and 50 μL of the fluorescent dye SYBR Green I was used for staining in a dilution of 1:10,000 in TE buffer (10 mM Tris-HCl, 1 mM EDTA, pH 7.5). The slides were observed using a fluorescent microscope (Olympus BX53, Tokyo, Japan) at 40× magnification.

Totally, 100 randomly selected cells (50 cells/experiment) were measured using the CaspLab—Comet Assay Software Project (version: casplab_1.2.3b2.exe) for the estimation of the tail parameters: Tail Moment (TM), Tail Length (TL), and % DNA Tail (TD). TM has been defined as the amount of DNA in the tail and the mean distance of migration in the tail [77].

### 4.4. Statistical Analysis

GraphPad Prism software version 8.0.1 was used (GraphPad Software, San Diego, CA, USA) for data statistical analysis by one-way ANOVA followed by Tukey’s analysis for XTT and Comet results and Mann-Whitney *t*-test for data from flow cytometry and TBARS methods. * *p* < 0.05; ** *p* < 0.01; *** *p* < 0.001; † *p* < 0.0001; n.s. (not significant) indicated the significance level.

## 5. Conclusions

HT is a well-studied component of olive tree and an approved food ingredient with multifaceted biological activity. For these reasons, the production of extracts rich in HT from natural sources is at the center of scientific interest because of the high demand on the market for products enriched with natural antioxidants. In the present study, a rapid methodology for the production of a HT-enriched extract from OLs was described. The proposed method was based on the direct acidic hydrolysis of powdered OLs, where the extraction procedure and the hydrolysis of OLE were carried out in one step. The techniques proposed in this study for HT-enriched extract production are widely used in the industry and can be easily scaled up. Furthermore, the bioactivity of this extract was investigated using cell-free and cell-based methods. Our results showed that the HT-enriched extract exerted significant in vitro antioxidant and geno-protective activity. These findings could be exploited in order for these extracts to be used as supplements in the pharmaceutical, food, and cosmetic industries.

## Figures and Tables

**Figure 1 molecules-29-00299-f001:**
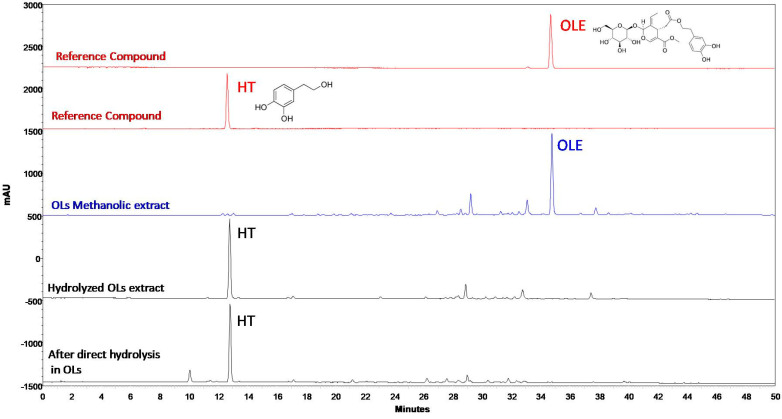
HPLC-DAD chromatograms of methanolic extract, extract originated from hydrolysis of methanolic extract (HOLE), and extract produced by the direct hydrolysis of dried OLs (DHOLE), applying the IOC proposed method. OLE and HT are highlighted.

**Figure 2 molecules-29-00299-f002:**
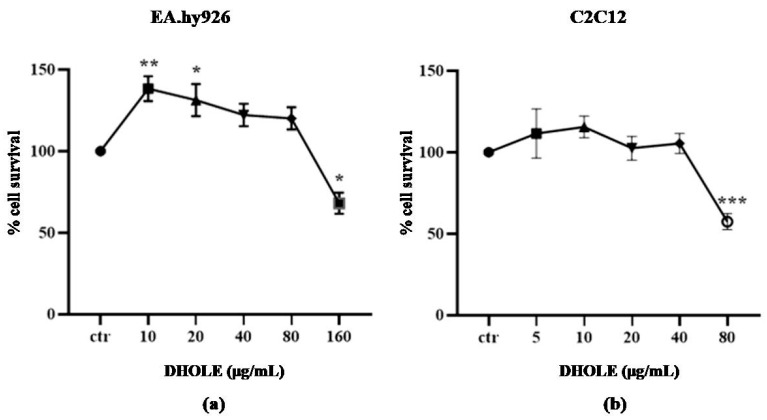
Estimation of non-cytotoxic concentrations of DHOLE at (**a**) EA.hy926 cells and (**b**) C2C12 cells. Cells were incubated in serum-free medium in the presence or absence (ctr) of increasing concentrations of DHOLE for 24 h, and an XTT assay was performed. The results represent the mean ± SEM of three independent experiments performed in triplicate and expressed as a % change from the control sample. * *p* < 0.05; ** *p* < 0.01; *** *p* < 0.001 indicated a statistically significant difference between the treated samples and the control.

**Figure 3 molecules-29-00299-f003:**
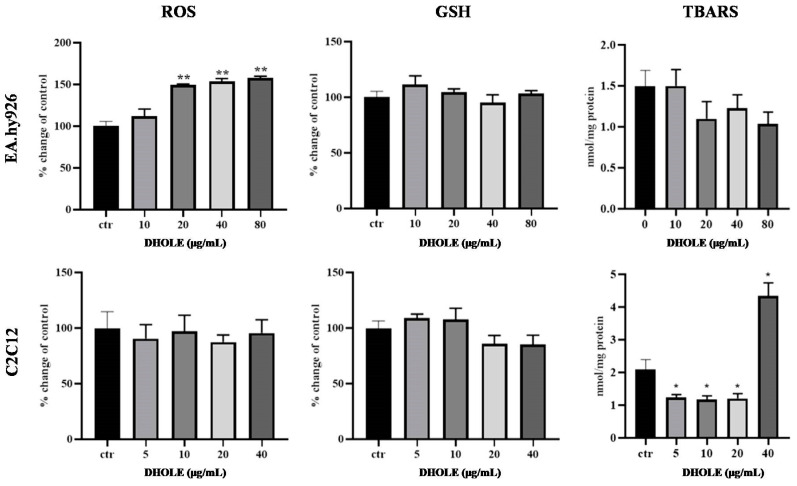
Antioxidant activity of DHOLE determined by measuring ROS, GSH, and TBARS levels. EA.hy926 and C2C12 cells were treated with increasing concentrations of DHOLE or not (ctr) for 24 h. All results are expressed as the mean ± SEM of at least three independent experiments. * *p* < 0.05; ** *p* < 0.01 indicated a statistically significant difference between the treated samples and the control.

**Figure 4 molecules-29-00299-f004:**
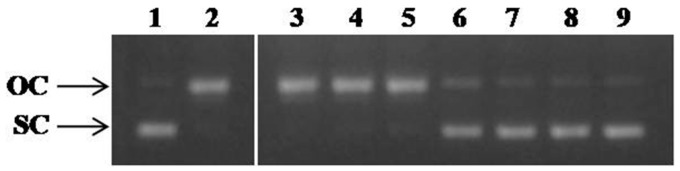
Protective activity on ROO^•^-induced oxidative damage. Lane 1: pBluescript-SK+ plasmid DNA without any treatment; Lane 2: plasmid DNA exposed to ROO^•^ alone; Lane 3–8: plasmid DNA exposed to ROO^•^ in the presence of increasing concentrations of DHOLE (Lane 3: 6.25 μg/mL; Lane 4: 12.5 μg/mL; Lane 5: 25 μg/mL; Lane 6: 50 μg/mL; Lane 7: 100 μg/mL; Lane 8: 200 μg/mL) Lane 9: plasmid DNA exposed to the maximum tested concentration of the extract alone (200 μg/mL). OC: open circular; SC: supercoiled (representative figure of the three experimental repeats; the two parts of the figure came from the same agarose gel, but among them were other samples).

**Figure 5 molecules-29-00299-f005:**
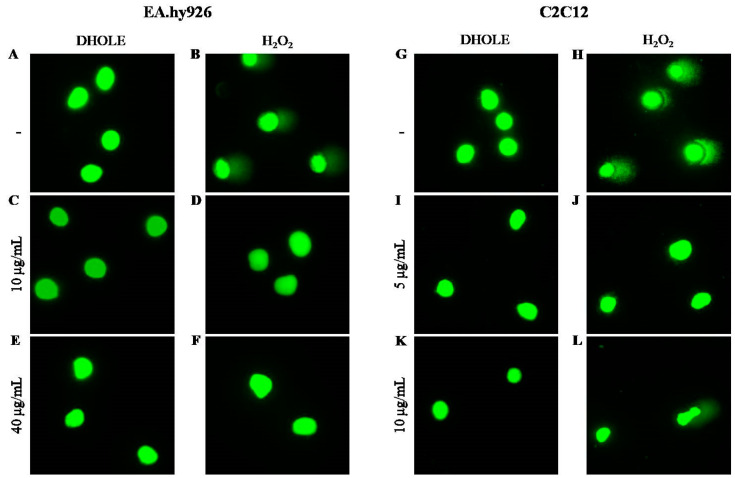
H_2_O_2_-induced DNA damage was prevented at lower concentrations of DHOLE. Representative images of the Comet assay after pre-treatment with DHOLE in the absence or presence of 250 μΜ H_2_O_2_ at EA.hy926 cells (**A**–**F**) and at C2C12 cells (**G**–**L**). (**A**,**G**): untreated cells (control), (**B**,**H**): cells treated with 250 μΜ H_2_O_2_ for 1 h, (**C**,**I**): treated cells with 10 μg/mL and 5 μg/mL, respectively, of DHOLE for 24 h, (**D**,**J**): pre-treated cells with 10 μg/mL and 5 μg/mL, respectively, of the extract for 23 h followed by incubation with 250 μΜ H_2_O_2_ for 1 h, (**E**,**K**): treated cells with 40 μg/mL and 10 μg/mL, respectively, of DHOLE for 24 h, (**F**,**L**): pre-treated cells with 40 μg/mL and 10 μg/mL, respectively, of the extract for 23 h followed by incubation with 250 μΜ H_2_O_2_ for 1 h.

**Figure 6 molecules-29-00299-f006:**
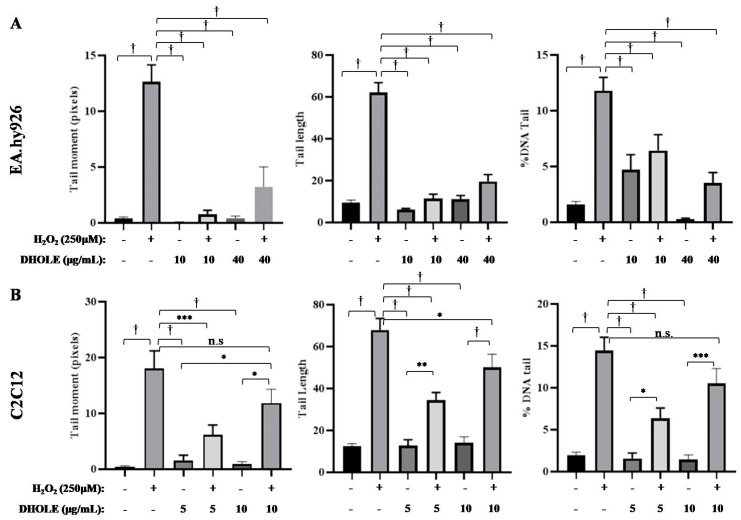
Results from the three parameters tail moment (TM), tail length (TL), and % DNA tail (TD) after automatic calculation of cell images using the CaspLab—Comet Assay Software Project (casplab_1.2.3b2.exe) at (**A**) EA.hy926 cells and (**B**) C2C12 cells. Cells were pre-treated at specific concentrations of DHOLE for 23 h: (**A**) 10 and 40 μg/mL for EA.hy926 cells and (**B**) 5 and 10 μg/mL for C2C12 cells. Then, 250 μM H_2_O_2_ was added or not for 1 h of treatment, and the next comet assay was performed. The data are presented as the mean ± SEM of 100 randomly selected cells. * *p* < 0.05; ** *p* < 0.01; *** *p* < 0.001; † *p* < 0.0001; n.s.: not significant indicated a statistically significant difference.

**Table 1 molecules-29-00299-t001:** HT and OLE quantification in the hydrolyzed and non-hydrolyzed OLs extracts. Data for the regression model (linear regression, r-squared, and concentration ranges) are given. The results were expressed in g of analyte per 100 g of extract.

Phenolic Compounds	g/100 g of OLs Methanolic Extract (Mean ± SD, n = 3)	g/100 g of HOLs Extract * (Mean ± SD, n = 3)	g/100 g of DHOLs Extract ** (Mean ± SD, n = 3)	Linearity of Phenolic Compound Standards		
				Linear Regression	r^2^	Concentration Range (μg/mL)
HT	˂LOQ ***	11.91	11.27	y = 87,594x + 31,144	0.999	0.6–100
OLE	9.36	˂LOD	˂LOD	y = 22,405x + 313,038	0.998	50–400

* After hydrolysis of the methanolic extract; ** After direct hydrolysis in olive leaves; *** LOQ below the limit of quantitation.

**Table 2 molecules-29-00299-t002:** IC_50_ and AU_0.5_ values (μg/mL) of DHOLE. Data are presented as the mean ± SD of two independent experiments, performed in triplicate.

	IC_50_ (μg/mL)					AU_0.5_ (μg/mL)
Samples	Solvent	DPPH^•^	ABTS^•+^	O_2_^•−^	ROO^•^	RP
DHOLE	H_2_O	21.3 ± 0.3	6.54 ± 0.8	161.0 ± 2.86	36.4 ± 2.22	7.9 ± 1.19
**Positive** **controls**						
Ascorbic acid	H_2_O	3.8 ± 0.1	2.8 ± 0.4	ND ^1^	290.0 ± 20.6	4.6 ± 0.3
Ellagic acid				260.0 ± 5.4		

^1^ ND: not detectable.

## Data Availability

Data are contained within the article and Appendix A.

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
