# Peer review of "The Bioactivity of a Hydroxytyrosol-Enriched Extract Originated after Direct Hydrolysis of Olive Leaves from Greek Cultivars"

_molecules, 2024, doi:10.3390/molecules29020299_

Round 1

Reviewer 1 Report

Comments and Suggestions for Authors

The authors carried out extensive and novel work on the hydrolysis of olive leaves and found good results, in addition to photoprotective activity. However, the manuscript must be improved in language and style. After that, it can be accepted with some other minor observations.

The structures of oleuropein and other compounds should be incorporated into the text.

In my opinion, the manuscript is difficult to read since there are grammatical and style errors. The entire manuscript must be reviewed by a native English-speaking expert.

In materials and methods, modify the subheadings, for example, “olive leaf extract” is so repetitive and obvious.

The conclusions are poor and must be improved.

Figure 1 should be enlarged, and the resolution improved.

References should be checked.

Comments on the Quality of English Language

Extensive editing of English language required

Reviewer 2 Report

Comments and Suggestions for Authors
    1. There is so much text without between-word spacing.  Please verify these sentences in the whole text of the article, mainly in the abstract section.

    2. Overall authors should give in 'results and discussions' their interpretations, discuss implications and limitations, and share their recommendations rather than summarizing key findings.

    3. I also suggest authors must improve their discussion of results by comparing their findings with those previously reported in the literature. 
    4. The authors should revise the Conclusion (lines 596-600) and try to state some perspectives regarding future studies and novelties improving their work rationale and stated novelties.

Comments on the Quality of English Language

Minor editing of English language required.

Reviewer 3 Report

Comments and Suggestions for Authors

I reviewed the article entitled "The bioactivity of an extract originated after direct hydrolysis of olive leaves from Greek cultivars", written by Maria Kourti, Zoi Skaperda, Fotis Tekos, Panagiotis Stathopoulos, Christina Koutra, Alexios Leandros Skaltsounis, and Demetrios Kouretas.

The purpose of this manuscript is to investigate the antioxidant and genoprotective effects of the hydroxytyrosol-enriched extract obtained by the direct hydrolysis of the dried leaves of the Olea europaea Greek cultivars.

The scientific collection is very interesting. However, some problems, as indicated below, should be addressed before the document can be considered for publication in this journal. This version of the manuscript is not enough complete. Here, I present all my objections in detail.

General remarks:

I suggest checking the style of the manuscript. Formatting text is not adequate, and several spaces are missing between words.

Title:

I suggest revising the title of the manuscript, and adding "hydroxytyrosol extract”.

Abstract:

Lines 19-22: Reformulate the concluding sentence, the meaning is unclear.

Introduction:

The introduction is very interesting, but it should be implemented.

Line 50: Use lowercase letters for compounds.

Line 59:  The strong antioxidant activity of hydroxytyrosol is not only displayed in the beneficial effects against several diseases. Recently, a strong protection against heavy metals-caused damages has been also clarified. In this regard, it might be interesting to add a short sentence, adding the following reference: doi: 10.3390/cells12030424; doi: 10.1007/s12010-018-2723-5.

Results:

·       Line 114: I suggest inserting other superscripts instead of "*", "**", and "***", which are used as symbols of statistical significance.

·       Lines 124-125: Reformulate the sentence.

·       Line 136, Table 2: Are the IC50 and AU0.5 values of HOLE available? Although the HT content of HOLE is DHOLE is similar, are the authors sure that the two extracts have the same antioxidant activity?

·       Line 171, Figure 3: I do not agree with expressing ROS, GSH, and TBARS levels as a percentage of change of control. The graph thus made could be misleading. I suggest indicating the actual percentages of these oxidative stress markers for each experimental condition.

·       Line 190, Figure 4: I suggest adding a histogram to highlight the obtained result.

·       Line 203: Has cell viability been verified following treatment with 250 µm H2O2 for 1h?

·       Line 231, Figure 6B: In the first and third graphs, the statistical significance between treatment with H2O2 and treatment with H2O2 + 10 µg/ml DHOLE is missing.

Discussion:

·       Line 272: Add “DHOLE” after “direct hydrolysis of OLE”.

Material and methods:

·       Line 379: “60oC” should be modified in “60°C”.

·       Line 382: Once the acronym is written, use it in the rest of the text (HT).

·       Line 387: Add the concentration of NaOH used.

·       Lines 378, and 399: Could the authors explain the difference between Protocol A and B in terms of powder weight (10 and 100 g respectively)?

·       Line 452: Check the units of measurement (µL, not µl).

·       Line 468: Write “PMS” in full.

·       Line 509: Were EA.hy926 cells grown in only 1 g/L of glucose?

References

The references of this work are few. I suggest implementing this section.

Comments on the Quality of English Language

English language and style are not exhaustive, a greater spell check is required to ensure that an international audience can clearly understand your text. In general, I suggest reviewing the style of the manuscript according to the guidelines of the journal.
